# Functional Characterization of Three GnRH Isoforms in Small Yellow Croaker *Larimichthys polyactis* Maintained in Captivity: Special Emphasis on Reproductive Dysfunction

**DOI:** 10.3390/biology11081200

**Published:** 2022-08-10

**Authors:** Zahid Parvez Sukhan, Yusin Cho, Shaharior Hossen, Seok-Woo Yang, Nam-Yong Hwang, Won Kyo Lee, Kang Hee Kho

**Affiliations:** 1Department of Fisheries Science, Chonnam National University, Yeosu 59626, Korea; 2Ocean and Fisheries Science Institute, Yeonggwan 59326, Korea

**Keywords:** GnRH, BPG axis, GtH, reproduction, reproductive dysfunction, small yellow croaker

## Abstract

**Simple Summary:**

Small yellow croaker is a popular marine fish in southeast Asian countries. Due to a decline in production in the wild, marine aquaculture has been initiated in Korea. Seed production is performed using captive-reared broodstock, which is known to undergo reproductive dysfunction. Reproductive dysfunction is closely linked to endocrinological dysfunction in the brain-pituitary-gonad (BPG) axis. Gonadotropin-releasing hormone (GnRH) acts as a central regulator in the BPG-axis. To determine the possible involvement of GnRHs in reproductive dysfunction of small yellow croaker reared in captivity, three GnRH isoforms were cloned and functional characterization has been performed. The expression of GnRH1 in the brain was significantly higher at the ripen stage in both sexes during gonadal development. Gonadotropin subunits (GPα, FSHβ, LHβ) were simultaneously increased in the pituitary at the ripen stage in both sexes. Interestingly, females showed significantly lower expression of GnRH1 and GtHs than males. Both in vivo and in vitro administration of GnRH1 showed that GtHs were increased significantly in the pituitary at high concentration. However, sex-steroids (E2 and MT) significantly inhibited the GnRH1 expression in the brain in a dose-dependent manner. Altogether, GnRH1 plays a key role in gonadal maturation, and a low level of GnRH1 and GtHs might be responsible for reproductive dysfunction in the female small yellow croaker.

**Abstract:**

Fish reproduction is regulated by the brain–pituitary–gonad (BPG) axis where the gonadotropin-releasing hormone (GnRH) plays a central role. Seed production of small yellow croaker (*Larimichthys polyactis*) is performed using captive-reared broodstock known to undergo reproductive dysfunction, which is connected to endocrinological dysfunction. To determine the endocrinological mechanism of GnRHs in the BPG axis of small yellow croaker, full-length sequences of three GnRH isoforms encoding sbGnRH (GnRH1), cGnRH-II (GnRH2), and sGnRH (GnRH3) were cloned and characterized from brain tissue. qRT-PCR, in vivo, and in vitro experiments were performed for functional characterization. The mRNA expression of GnRH1 in the brain and gonadotropin subunits (GPα, FSHβ, and LHβ) in the pituitary were significantly higher at the ripen stage during gonadal development and GnRH1 at spawning stage during spawning events. Expression of both GnRH1 and GtH subunits was significantly lower in females than males. GtH subunits were induced at higher concentrations of GnRH1 in vivo and in vitro. Sex-steroids significantly inhibited the GnRH1 expression in vitro in a dose-dependent manner. Taken together, results indicated that GnRH1 plays a key role in gonadal maturation and sex-steroids induced negative feedback in the regulation of GnRH. A lower level of GnRH1 and GtHs might be responsible for reproductive dysfunction in a female small yellow croaker.

## 1. Introduction

Reproduction in fish is mainly regulated by the integration of signals of endogenous neuroendocrine genes and hormones with exogenous environmental factors [1]. The brain–pituitary–gonad (BPG) axis is a key neuroendocrine system involved in the reproductive processes in vertebrates including fish [2]. Brain gonadotropin-releasing hormone (GnRH) acts as a key upstream regulator in the BPG axis. It stimulates the synthesis and release of gonadotropins (GtHs), which then act on the gonads to synthesize sex steroids to regulate gonadal maturation and reproduction. Consequently, the steroids send feed back to the brain and pituitary to compete the BPG axis [2]. Vertebrate GnRHs are 10 amino acid long neuropeptides. To date, a number of GnRH isoforms have been identified in both vertebrate and invertebrate species, eight of them have been identified in fish [3]. It is evident that three or two GnRH isoforms are present in a single vertebrate species and classified as GnRH1, GnRH2, and GnRH3 [4]. In this classification, chicken GnRH-II (cGnRH-II) and salmon GnRH (sGnRH) are termed as GnRH2 and GnRH3, respectively, while most other GnRH isoforms are termed as GnRH1.

GnRH1 has been found in all teleost species that possess three GnRH isoforms except those possessing two GnRH isoforms such as sockeye salmon and zebrafish [3]. This form of GnRH is known as veritable GnRH that is mainly localized in the telencephalon of the preoptic area in the brain. It regulates gonadal maturation and reproduction by stimulating pituitary gonadotropins. GnRH2 is localized in the midbrain tegmentum which is highly conserved in all forms of vertebrate species. It appears to be a neuromodulator and/or neurotransmitter, and it could modulate reproductive behavior [5], food intake and energy balance [6,7]. GnRH3 is localized in the olfactory bulb and considered as a teleost specific GnRH isoform. It has been shown that GnRH3 has neuromodulatory function, and it can regulate sexual behavior, including spawning migration [8].

It has been suggested that three GnRH isoforms represent three paralogous genes arising from an ancestral GnRH via two rounds (2R) of whole genome duplication (WGD, tetraploidization) that seems to be materialized during the early stage of vertebrate evolution. This genome duplication event is also followed by two separate deletion events [3,9,10]. In line with this hypothesis, it is understood that these three GnRH isoforms are already retained in the common ancestor of all sorts of jawed vertebrate lineages.

The synthesis and secretion of gonadotropin (GtH) subunits (glycoprotein α, GPα; follicle-stimulating hormone β, FSHβ; and luteinizing hormone β, LHβ) in the pituitary are generally regulated by the stimulation from GnRH. GtHs are crucial in the regulation of gonadal maturation in fish [11]. Stimulatory effects of GnRH on GtH subunits have been studied in a few fish species so far, with results showing that GnRH can upregulate the expression of GtHs in striped bass [12], black porgy [13], spotted scat [14], and pompano [15]. Then, sex steroids are secreted in the gonads by the stimulation of GtHs, which are responsible for gonadal maturation and reproduction [2]. Consequently, steroids give feedback to the brain and pituitary to maintain homeostatic regulation of reproductive function and behavior that could potentially inhibit the synthesis and/or secretion of GnRH from the brain [16]. Several studies have been performed on the feedback regulation of steroids such as estradiol (E2), progesterone (P), and testosterone (T) on GnRHs in BPG axis including mammals and fish [14,15]. Sex steroid 17β-estradiol (E2) and 17α-testosterone (MT) can regulate both the expression and secretion of GnRHs [14,17,18]. Both positive and negative effects of E2 and MT on GnRH expression in fish have been reported in several fish species [14,15,19]. Therefore, it is important to study the effects of sex steroids on GnRHs in experimental fish and determine whether it could mediate a positive or negative feedback.

The small yellow croaker, *Larimichthys polyactis*, is a marine demersal oceanodromous fish that belongs to the family Sciaenidae and order Perciformes. It is mainly distributed in the west coast of South Korea, Bohai Sea, and East China Sea [20]. Due to its high nutritional value and superior meat quality, it is a popular and commercially important marine fish in Korea. However, its wild population is depleted severely due to overfishing and deterioration of its native habitat [21]. To meet the increasing demand of this fish, marine farming has been initiated in southeast Asian countries including Korea. Recently, artificial breeding has also been developed for this species using captive-reared broodstock in Korea and China [22]. However, captive-reared small yellow croaker, particularly female, failed to undergo final gonadal maturation and spawning [23]. It has been reported that fish raised in captivity show three types of reproductive dysfunctions; (a) fish completely fail to undergo vitellogenesis and spermatogenesis, (b) absence of final oocyte maturation, and (c) absence of spawning at the end of reproductive cycle [24]. Generally, these kinds of reproductive dysfunctions occurred due to the absence of the appropriate spawning environment in captivity and are closely associated with endocrinological dysfunction related to the BPG axis [2,24,25]. Endocrinological dysfunctions in captive-reared fish have been reported in several fish, including greater amberjack [25], turbot [26], jack mackerel [27], and long-whiskered catfish [28]. To overcome the endocrinological dysfunction in captive-reared fish broodstock, it is important to understand the detailed endocrinological function in the BPG axis of a particular fish species. Although several studies have been reported the reproductive status and breeding of small yellow croaker [29,30,31], neuroendocrinological studies associated with GnRHs have not been published yet. To understand functional involvement of GnRHs related to gonadal maturation and reproductive dysfunction in the BPG axis of small yellow croaker reared in captivity, three GnRHs isoforms were cloned and characterized in the present study. Furthermore, mRNA expression levels of GnRHs in the brain and expression levels of GtHs in the pituitary at different reproductive and spawning stages were observed. In vivo and in vitro effects of GnRH on GtHs in the pituitary were also observed. Finally, feedback regulation of sex steroids (E2 and MT) on GnRHs was evaluated by in vitro study. The evolutionary relationship of three GnRH isoforms was also studied by phylogenetic and synteny analyses.

## 2. Materials and Methods

### 2.1. Experimental Fish and Rearing Condition

Adults of small yellow croaker at the gonadal recovery phase were collected from the Yellow Sea, transported, and reared in circular tank with running sea water in the Ocean & Fisheries Science Institute, Yeonggwang-gun, Jeollanam-do, South Korea. Fish were maintained under natural daylight conditions and fed with small shrimp, oyster, and squid twice per day, later supplemented with pellet food. Fish were then transferred to a broodstock conditioning tank where water temperature was maintained initially at 11 °C, slowly raised to 18 °C (△1 °C every 7 days), and maintained at 18 °C until spawning. Oxygen was supplied continuously in the rearing tank.

### 2.2. Tissue Collection for Gene Cloning

Adult fish of both sexes were collected from rearing tanks of Ocean & Fisheries Science Institute, Jeollanam-do and transported to the Laboratory of Molecular Physiology, Department of Fisheries Science, Chonnam National University, South Korea. All fish were anesthetized with tricaine methanesulfonate (MS222) prior to collection of tissue samples. Brain samples were collected for cloning and isolation of GnRH genes. Collected brain samples were washed with 10 mM phosphate buffered saline (PBS), immediately snap frozen in liquid nitrogen, and stored at −80 °C until extraction of total RNA.

### 2.3. Tissue Collection for mRNA Expression Analysis from Different Experimental Conditions

All tissue samples were collected after anesthetizing fish with MS222. All collected tissue samples were washed with 10 mM PBS, immediately snap frozen in liquid nitrogen, and stored at −80 °C until total RNA extraction.

#### 2.3.1. Different Organ Tissues of Adult Small Yellow Croaker of Both Sexes

A total of 10 adult fish of both sexes were euthanized to collect tissue samples of different organs. Collected tissues were brain (BRN), pituitary (PIT), gill (GIL), liver (LIV), heart (HRT), kidney (KID), muscle (MUS), ovary (OVR), and testis (TES).

#### 2.3.2. Different Brain Parts of Adult Small Yellow Croaker in Both Sexes

Brain samples from 10 fully mature fish of both sexes were collected. Brains were then dissected to collect three different areas: [A] olfactory bulbs, [B] telencephalon and preoptic area, and [C] mid brain (cerebellum, caudal optic tectum, and thalamus).

#### 2.3.3. Brain and Pituitary at Different Gonadal Developmental Stages in Both Sexes

Brain and pituitary samples of small yellow croaker of both sexes at four gonadal developmental stages were collected. Gonadal developmental stages of small yellow croaker determined as described previously [30]. Developmental stages were included immature (IM), developing stage (DS), ripe stage (RS), and spent stage (SS).

#### 2.3.4. Brain of Small Yellow Croaker during Induced Spawning Event in Both Sexes

Brain samples were collected during different induced spawning events of small yellow croaker of both sexes. Small yellow croaker at gonadal ripen stage were injected with GnRHa (Sigma, St. Louis, MO, USA) at a dose of 10 μg/100 g-bw (gram-body weight) for final gonadal maturation and reared in a circular tank with running seawater and continuous oxygen supply. Release of gametes was started after 24 h of injection. Brain tissues of both male and female fish were collected at three different time-points: before injection of GnRHa termed as before spawning (BSW), after 24 h of injection or during release of gametes or spawning (DSW) and at 7 days after releasing gametes or post-spawning (PSW).

#### 2.3.5. Pituitary of Small Yellow Croaker from In Vivo Induction of GnRH1 Peptide

Pituitaries from female small yellow croaker were collected after administration of GnRH1 (sbGnRH) peptide as described previously [14,15] with slight modifications. Briefly, GnRH1 peptide (Sigma, USA) was dissolved in 10 mM PBS. Thirty adult small yellow croakers at the gonadal ripen stage were interperitoneally injected in three different concentrations 1, 10, and 100 ng/g-bw. In each treatment group, 10 fish were injected with GnRH1 peptide and reared in a circular tank with continuous water supply and aeration. For the control group, 10 mM PBS was injected intraperitoneally. After injection, pituitaries were collected at 3 h, 6 h, and 12 h post-injection and stored at −80 °C.

#### 2.3.6. Pituitary of Small Yellow Croaker from In Vitro Incubation with GnRH1 Peptide

Pituitaries of female small yellow croaker were incubated with GnRH1 peptide (Sigma, St. Louis, MO, USA) in vitro as described previously [14,15] with slight modifications. Briefly, pituitaries from 30 mature fish were collected, washed with M199 medium modified with Hanks’ Balanced Salts (Gibco, USA), and transferred to 12-well culture plates, three pituitaries per well. Pituitaries were pre-incubated with serum-free M199 medium (modified with Hanks’ balanced salts) treated with penicillin–streptomycin for 2 h at 25 °C. After pre-incubation, the medium was discarded. Pituitaries were then washed with M199 medium twice and then incubated with 0.1, 1, and 10 µM GnRH1 peptide in M199 medium for 12 h at 25 °C. A control group was incubated without GnRH1 peptide. After incubation, pituitaries were collected at 3 h, 6 h, and 12 h, and stored at −80 °C until RNA extraction.

#### 2.3.7. Hypothalamus of Small Yellow Croaker from In Vitro Incubation with E2 and MT

To know the feedback effect of steroids on GnRH, hypothalamuses of female small yellow croaker were incubated with 17β-Estradiol (E2) or 17α-Methyltestosterone (MT) in vitro as described previously [14,15] with slight modifications. Briefly, E2 and MT (Sigma, USA) were first dissolved in acetone and absolute ethanol, respectively. Hypothalamuses were collected from 30 reproductively mature female fish and washed with M199 medium modified with Hanks’ Balanced Salts (Gibco, Grand Island, NY, USA), transferred to 12-well culture plates, three hypothalamuses in a well and pre-incubated with M199 medium (modified with Hanks’ balanced salts) treated with penicillin–streptomycin at 25 °C for 2 h. After pre-incubation, the medium was discarded, and hypothalamuses were washed twice with the same medium. Hypothalamuses were then incubated with M199 medium containing 0.1, 1, and 10 μM E2 or MT for 12 h. A control group was incubated without E2 or MT. After incubation, hypothalamuses were collected at 3 h, 6 h, and 12 h, and stored at −80 °C.

### 2.4. RNA Extraction and cDNA Syntheses

An ISOSPIN Cell and Tissue RNA kit (Nippon Gene, Tokyo, Japan) was used to extract total cellular RNAs from all sampled tissues. Syntheses of first strand cDNAs were performed using oligo(dT) primer (OdT) (Sigma) and a superscript III First-strand cDNA synthesis kit (Invitrogen, Carlsbad, CA, USA) from 1–4 μL of total RNAs. In addition, 5′- and 3′-RACE cDNAs were synthesized from 1 μL of whole brain total RNA using a SMARTer^®^ RACE 5′/3′ Kit (Takara Bio Inc., Kusatsu, Shiga, Japan). All steps of RNA extraction and cDNA synthesis were conducted following the manufacturer’s protocol.

### 2.5. Cloning and Sequencing of Three Full-Length GnRH Isoforms in Small Yellow Croaker

#### 2.5.1. Cloning of Partial Sequences

To obtain partial fragment of seabream GnRH (sbGnRH; GnRH1), chicken GnRH-II (cGnRH-II; GnRH2), and salmon GnRH (sGnRH; GnRH3) genes, the reverse transcription polymerase chain reaction (RT-PCR) was performed using a whole brain cDNA templet. Primer sets (forward and reverse) were designed from nucleotide sequences of known GnRH isoforms of Atlantic croaker, *Micropogonias undulatus* (sbGnRH, GenBank Accession No. AY324668.2; cGnRH-II, GenBank Accession No. AY324669.2; and sGnRH, GenBank Accession No. AY324670.2). All primers used in cDNA cloning are presented in Appendix A. The cloning and sequencing of partial nucleotide fragment of three GnRH isoforms were carried out following the protocol as described previously [32]. The RT-PCR thermal cycling condition was initial denaturation at 95 °C for 2 min; followed by 35 cycles of denaturation at 95 °C for 1 min, annealing at 58 °C for 1 min, and extension at 72 °C for 1 min; and a final extension step at 72 °C for 5 min.

#### 2.5.2. Cloning of 5′- and 3′-RACE Sequences

Rapid amplification of cDNA ends (RACE) PCR was performed to obtain full-length sequences of three GnRH isoforms. The cloning and sequencing of 5′- and 3′-RACE fragments of three GnRH isoforms were carried out using a previously described protocol [33] and as recommended by the SMARTer^®^ RACE 5′/3′ kit (Takara Bio Inc., Kusatsu, Shiga, Japan). Gene-specific 5′- and 3′-RACE primes (Appendix A) were designed from cloned partial nucleotide sequence of each GnRH isoforms as recommended in SMARTer^®^ RACE 5′/3′ kit. RACE PCR was carried out with 35 cycles for both 3′-RACE and 5′-RACE for each gene. Thermal cycle conditions were maintained as prescribed in the kit. Finally, sequences of 5′- and 3′-RACE fragments were combined, and overlaps with the initially cloned partial cDNA fragment were trimmed to obtain the full-length sequences of GnRH1, GnRH2, and GnRH3 genes.

### 2.6. In-Silico Analysis of Sequences of the Three Cloned GnRH Isoforms

Several online tools and software were used to analyze nucleotide and amino acid sequences of the three cloned GnRH isoforms. Deduced amino acid sequences of each gene was generated from cloned full-length nucleotide sequences using the EMBOSS Transeq (http://www.ebi.ac.uk/Tools/st/emboss_transeq/; accessed on 3 April 2022) online tool. ORFfinder (https://www. ncbi.nlm.nih.gov/orffinder/; accessed on 3 April 2022) was used to predict open reading frames (ORFs) and potential protein encoding segments. Protein homology was analyzed using the Basic Local Alignment Search Tool (BLASTP; https://blast.ncbi.nlm.nih.gov/Blast.cgi?PAGE=Proteins; accessed on 3 April 2022). Molecular weight and theoretical isoelectric point (pI) of proteins were computed using ProtParam (https://web.expasy.org/protparam/; accessed on 3 April 2022) and Protcomp 9.0 (http://www.softberry.com/berry.phtml; accessed on 3 April 2022), respectively. Gene ontology (GO) of three GnRH proteins was predicted using an online tool, the Contact-guided Iterative Threading ASSEmbly Refinement (C-I-TASSER) protein structure prediction server (https://zhanggroup.org/C-I-TASSER/; accessed on 4 April 2022). Functional domains of proteins were determined using the SMART (http://smart.embl-heidelberg.de/; accessed on 3 April 2022) or NCBI conserved domain search tool (http://www.ncbi.nlm.nih.gov/Structure/cdd/wrpsb.cgi; accessed on 3 April 2022). Conserved motifs in the three GnRHs amino acid sequences were discovered using Multiple Em for Motif Elicitation (MEME) online tools v.5.4.1 (http://meme-suite.org/tools/meme; accessed on 4 April 2022). Sequence information of GnRHs used for motif analysis of the three GnRH isoforms are presented in Appendix A.

### 2.7. Analysis of Amino Acid Sequence Alignment and Identity-Similarity Index

Representative amino acid sequences of GnRHs were obtained from the NCBI protein database (https://www.ncbi.nlm.nih.gov/protein/; accessed on 5 April 2022). Amino acid sequences of three GnRH isoforms from different fish were aligned using a multiple sequence alignment program, ClustalOmega (https://www.ebi.ac.uk/Tools/msa/clustalo/; accessed on 5 April 2022). The alignment of protein sequences was edited and visualized using Jalview ver. 2.11.1.7 software. Amino acid sequence identity and similarity of the three GnRH isoforms of small yellow croaker with those of GnRH isoforms of other teleost species were calculated using “Idnt & Sim” online suite (https://www.bioinformatics.org/sms2/ident_sim.html; accessed on 7 April 2022). First, amino acid sequences of corresponding GnRHs were obtained from the NCBI protein database, aligned with uniport online align tool (https://www.uniprot.org/align/; accessed on 7 April 2022) and finally analyzed with the “Idnt & Sim” suite against small yellow croaker GnRH isoforms. GnRH amino acid sequences used for multiple sequence alignment analysis are listed in Appendix A.

### 2.8. Prediction of 3D Protein Structure of Three GnRH Isoforms and Their GAP Regions

Three-dimensional (3D) structures of three GnRH isoforms and their GnRH-associated peptide (GAP) region proteins were modeled using an online protein structure and functional prediction program, I-TASSER (Iterative Threading ASSEmbly Refinement) server (https://zhanglab.ccmb.med.umich.edu/I-TASSER/; accessed on 4 April 2022). GAP sequences were limited between the dibasic site of proteolytic processing after the GnRH decapeptide and the stop codon of the ORF. Predicted 3D structures were visualized and analyzed using UCSF ChimeraX software v. 1.2.5 (https://www.cgl.ucsf.edu/chimera/; accessed on 8 April 2022).

### 2.9. Phylogenetic Analysis

Protein sequences of 47 different GnRH precursors encoding GnRH1, GnRH2, and GnRH3 of teleost fish were obtained from the NCBI protein database. Phylogenetic analysis was conducted with an alignment build option of MEGA software (ver.11.0.8; Sudhir Kumar, Philadelphia, PA, USA). Proteins were aligned using the MUSCLE alignment option. A phylogenetic tree was constructed using a neighbor-joining algorithm of the bootstrap method in MEGA software (v.11). Numbers shown on branches indicate the significance of nodes based on 10,000 bootstrap replications’ analyses. A detailed list of GnRH amino acid sequences used for phylogenetic analysis is presented in Appendix A.

### 2.10. Synteny Analysis of Three GnRH Isoforms of Small Yellow Croaker

Three synteny maps were generated for the three GnRH isoforms using a web-based synteny browser Genomicus ver. 106.01 (https://www.genomicus.bio.ens.psl.eu/genomicus-106.01/cgi-bin/search.pl; accessed on 12 April 2022). Three GnRH isoforms of small yellow croaker were identified by searching flanking sequences using NCBI BLASTX against a small yellow croaker genome database (Genome assembly: ZAAS_Lpoly_1.0; GenBank GCA_018985215.1). Genes flanking the GnRH isoforms in five chordate species viz. large yellow croaker (*Larimichthys crocea*), yellowtail amberjack (*Seriola lalandi*), fugu (*Takifugu rubripes*), Japanese medaka (*Oryzias letipes*), and human (*Homo sapiens*) were obtained from Genomicus as mentioned previously [34].

### 2.11. Quantitative Real-Time PCR (qRT-PCR) Analysis

To quantify relative mRNA expressions of three GnRH isoforms and three GtH subunits in different tissues of small yellow croaker, qRT-PCR analysis was performed. Expression levels of three GnRH isoforms were observed in different organs of small yellow croaker, three different brain areas, brains at different gonadal developmental stages (IM, DS, RS, and SS), brains during induced spawning events, hypothalamus from in vitro incubation with E2 or MT. The mRNA expression of three GtH subunits were observed in the pituitary at different gonadal developmental stages, after in vivo administration of GnRH1 peptide, and in vitro incubation with GnRH1 peptide.

All qRT-PCR assays were conducted using a 2 × qPCRBIO SyGreen Mix Lo-Rox kit (PCR Biosystems Ltd., London, UK) as described previously [35]. Each qRT-PCR reaction mixture was prepared in a total volume of 20 μL containing cDNA template (1 μL), 10 pmol gene specific forward and reverse primer (1 μL each), SyGreen Mix (10 μL), and double distilled water (10 μL). Triplicate reactions were performed for target and reference genes in each tissue sample. PCR amplification conditions were: preincubation at 95 °C for 2 min, followed by 40 cycles of a three-step amplification at 95 °C for 30 min, 60 °C for 20 s, and 72 °C for 30 s. The melting temperature was set as the instrument default setting. At the end of each cycle, a fluorescence reading was recorded for quantification. A LightCycler^®^ 96 System (Roche, Mannheim, Germany) was used for amplification and data analysis. The relative gene expression was determined using the 2^−ΔΔCT^ method with small yellow croaker *β-actin* gene as an internal reference. All primers used in qRT-PCR analysis are presented in Appendix A.

### 2.12. Statistical Analysis

Values of mRNA expressions of GnRHs in different organs, different brain part, brains in gonadal developmental stages, brains in spawning events, and mRNA expressions of GtH subunits in gonadal developmental stages were analyzed statistically and expressed as mean ± standard deviation (SD). Changes in relative mRNA expression of GnRHs and GtHs in different samples were analyzed by nonparametric one-way analysis of variance (ANOVA) using GraphPad Prism software (ver. 9.4.1; GraphPad Software LLC., San Diego, CA, USA). Tukey’s post hoc test was performed to assess statistically significant differences among different experimental tissues. Statistical significance was set at *p* < 0.05. Levels of mRNA expression of GtH subunits in time- and dose-dependent in vivo and in vitro experiments, and GnRH isoforms in time- and dose-dependent in vitro effect of E2 or MT experiment were analyzed by nonparametric one-way ANOVA and statistical significance were compared to zero time point initial control. All graphs were prepared using GraphPad Prism 9.3.1 software. Different letters or asterisks on bars or lines in figures indicate significant differences (*p* < 0.05).

## 3. Results

### 3.1. Cloning, Characterization, and In-Silico Analysis of Three GnRH Isoforms

Full-length cDNA sequences of small yellow croaker sbGnRH (GnRH1), cGnRH-II (GnRH2), and sGnRH (GnRH3) were cloned and sequenced from brain tissue (Figure 1).

#### 3.1.1. Molecular Characterization and In-Silico Analysis of sbGnRH (GnRH1) Isoform

Full-length sequences of small yellow croaker GnRH1, referred to syc-GnRH1 in this study (GenBank accession No. OK042349), were 380 bp long including a poly-A tail (Figure 1A). 5′- and 3′-untranslated regions (UTR) were 47 bp and 60 bp long, respectively. A putative polyadenylation signal (AATAAA) was recognized at 18 bp upstream of poly-A tail. The open reading frame (ORF) of the syc-GnRH1 cDNA sequence was 270 bp, encoding a putative protein with 90 deduced amino acids (GenBank Protein ID: UFT26658.1). Domain architecture analysis revealed a signal peptide (1–23 aa), GnRH1 decapeptide (24–33 aa), a proteolytic cleavage site (34–36 aa), and a GnRH1-associated peptide (37–90 aa). Motifs of syc-GnRH1 were analogously expressed when it was compared with different sbGnRH protein sequences with motif widths ranging from 11–29 amino acids. A total of four motifs were recognized in syc-GnRH1. Similarly, four motifs were recognized in other sbGnRH proteins compared in this study (Appendix A).

Theoretical molecular weight and isoelectric point (pI) of syc-GnRH1 protein were 10.08 kDa and 7.65 respectively. The aliphatic index of the protein was 69.22. The instability index was computed as 43.72, classifying the protein as unstable. Neural nets-nuclear prediction and integral prediction of protein location scores were 2.7 and 10.0, respectively, which predicted the protein as an extracellular (secreted) protein. Gene ontology (GO) term analysis predicted the syc-GnRH1 protein as a gonadotropin hormone-releasing hormone activity protein (GO:0005183) in molecular function with a C-scoreGO of 0.83 (Appendix A), regulation of cellular process protein (GO:0044767) in biological process with a C-scoreGO of 0.53 (Appendix A), and an extracellular region (GO:0005576) intracellular part protein (GO:0044424) in a cellular component with a C-scoreGO of 0.67 and 0.52, respectively (Appendix A).

#### 3.1.2. Molecular Characterization and In-Silico Analysis of cGnRH-II (GnRH2) Isoform

Full-length sequences of small yellow croaker GnRH2, referred to syc-GnRH2 in this study (GenBank accession No. OK042350), was 636 bp long including a poly-A tail (Figure 1B). 5′- and 3′-untranslated regions (UTR) were 144 bp and 232 bp, respectively. A putative polyadenylation signal (AATAAA) at 19 bp upstream of poly-A tail was recognized. The ORF of the syc-GnRH2 cDNA sequence was 255 bp, encoding a putative protein with 85 deduced amino acids (GenBank Protein ID: UFT26659.1). Domain architecture analysis revealed that it had a signal peptide (1–23 aa), GnRH2 decapeptide (24–33 aa), a proteolytic cleavage site (34–36 aa), and a GnRH2-associated peptide (37–85 aa). Motifs of syc-GnRH2 were analogously expressed when it was compared with GnRH2 protein sequences of different fish with motif width ranging from 11 to 49 amino acids. A total of three motifs were recognized in syc-GnRH2. Similarly, three motifs were also recognized in other GnRH2 proteins compared in this study (Appendix A).

Theoretical molecular weight and isoelectric point (pI) of syc-GnRH2 protein were 9.55 kDa and 8.84, respectively. Aliphatic index of the protein was 106.71. Instability index (II) was computed as 60.37, classifying the protein as an unstable protein. Neural nets-nuclear prediction and integral prediction of protein location scores were 3.0 and 10.0, respectively, which predicted the protein as an extracellular (secreted) protein. Gene ontology (GO) term predicted syc-GnRH2 protein as a gonadotropin hormone-releasing hormone activity protein (GO:0005183) in molecular function with a C-scoreGO of 0.80 (Appendix A), single-organism development process protein (GO:0044767) in a biological process with a C-scoreGO of 0.56 (Appendix A), and a plasma membrane protein (GO:0005576) in a cellular component with a C-scoreGO of 0.75 (Appendix A).

#### 3.1.3. Molecular Characterization and In-Silico Analysis of sGnRH (GnRH3) Isoform

A full-length sequence of small yellow croaker GnRH3, referred to syc-GnRH3 in the present study (GenBank accession No. OK042351), was 519 bp including a poly-A tail (Figure 1C). 5′- and 3′-untranslated regions (UTR) of the sequence were 29 bp and 217 bp, respectively. Two putative polyadenylation signals (AATAAA) were recognized at 14 bp and 175 bp upstream of poly-A tail. The ORF of the syc-GnRH3 cDNA sequence was 270 bp, encoding a putative protein with 90 deduced amino acids (GenBank Protein ID: UFT26660.1). Domain architecture analysis revealed a signal peptide (1–23 aa), GnRH3 decapeptide (24–33 aa), a proteolytic cleavage site (34–36 aa), and a GnRH3-associated peptide (37–90 aa). Motifs of syc-GnRH3 protein were analogously expressed when it was compared with GnRH3 protein sequences of different fish species with a motif ranging from 8 to 50 amino acids. A total of three motifs were recognized in syc-GnRH3. Three motifs were also recognized in other compared GnRH3 proteins (Appendix A).

Theoretical molecular weight and isoelectric point (pI) of sycGnRH3 protein were 10.04 kDa and 8.18, respectively. The aliphatic index of the protein was 92.00. Instability index (II) was computed as 42.88, classifying the protein as unstable. Its neural nets-nuclear prediction and integral prediction of protein location scores were 1.9 and 10.0, respectively, which predicted the protein as an extracellular (secreted) protein. Gene ontology (GO) term analysis predicted the sycGnRH3 protein as a gonadotropin hormone-releasing hormone activity protein (GO:0005183) in molecular function with a C-scoreGO of 0.87 (Appendix A), a single-organism development process (GO:0044767) protein in biological process with a C-scoreGO of 0.63 (Appendix A), and a cell part (GO:0044464) extracellular region (GO:0005576) protein in a cellular component with a C-scoreGO of 1.00 and 0.70, respectively (Appendix A).

### 3.2. Sequence Alignment and Identity–Similarity Index

Results of amino acid sequence alignment of small yellow croaker GnRH1, GnRH2, and GnRH3 with GnRHs amino acid sequences of other representative teleost species are presented in Figure 2. Sequence alignment revealed that GnRH decapeptide positioned at 24–33 aa was conserved in all three GnRH isoforms, and those aligned with other related GnRHs amino acid sequences. The identities and similarities among the deduced amino acid sequences of three GnRH isoforms of small yellow croaker and GnRHs of other representative teleost species are presented in Appendix A. Small yellow croaker GnRH1 showed the highest identity and similarity with GnRH1 of large yellow croaker (97.78% and 98.89%) and Atlantic croaker (96.67% and 98.89%). Small yellow croaker GnRH2 and GnRH3 also showed the highest identity and similarity with GnRH2 and GnRH3 isoform of large yellow croaker (100.00% and 100.00%, 98.82% and 100.00%, respectively) and Atlantic croaker (97.78% and 98.89%, 95.56%, and 97.78%, respectively).

### 3.3. Analysis of 3D Protein Structure of Three GnRH Isoforms and Their GAP Region

Three-dimensional (3D) secondary protein structures of three GnRH isoforms and three GnRH-associated peptides (GAP) variants were predicted using the I-TASSER server and presented in Figure 3. 3D structures of GnRH isoforms were characterized by several alpha helices separated by loops, forming a helix–loop–helix (HLH) structure. The 3D structure of syc-GnRH1 characterized by three alpha helices was separated by two loops (Figure 3A) with a C-score of −3.78, an estimated TM-score of 0.30 ± 0.10, and an estimated root-mean-square deviation (RMSD) of 12.3 ± 4.4 Å. On the other hand, the 3D-structure of syc-GnRH2 was characterized by two alpha helices separated by a loop (Figure 3B) with a C-score of −3.31, an estimated TM-score of 0.35 ± 0.12, and an estimated RMSD of 10.9 ± 4.4 Å. In contrast, the 3D-structure of syc-GnRH3 represented by four alpha helices was separated by three loops (Figure 3C) with a C-score of −3.26, an estimated TM-score of 0.32 ± 0.11, and an estimated RMSD of 11.8 ± 4.5 Å.

3D-structures of GAP sequences of three syc-GnRH isoforms were also predicted to have a classical helix–loop–helix (HLH) structure. The predicted 3D model of syc-GAP1 (small yellow croaker GnRH1-associated peptide) was characterized by two alpha helices separated by a loop (Figure 3D) with a C-score of −2.43, an estimated TM-score of 0.43 ± 0.14, and an estimated RMSD of 7.8 ± 4.4 Å. Similarly, the 3D model of syc-GAP2 was characterized by two alpha helices separated by a loop (Figure 3E) with a C-score of −1.75, an estimated TM-score of 0.50 ± 0.15, and an estimated RMSD of 6.1 ± 3.7 Å. On the other hand, the 3D structure of syc-GAP3 had no classical HLH structure. It showed a single alpha helix surrounded by two loops (Figure 3F) with a C-score of −2.93, an estimated TM-score of 0.38 ± 0.13, and an estimated RMSD of 8.9 ± 4.6 Å.

### 3.4. Phylogenetic Analysis

A phylogenetic tree was constructed using neighbor-joining method to assess three GnRH isoforms of small yellow croaker and their possible evolutionary connections with GnRHs in other representative fish species. An unrooted phylogenetic tree based on GnRHs amino acid sequences from fish showed that GnRH1, GnRH2, and GnRH3 were clustered in three separate clades (Figure 4). Small yellow croaker sbGnRH isoform was fitted with the GnRH1 clade, whereas small yellow croaker cGnRH-II and sGnRH isoforms were fitted with GnRH2 and GnRH3 clades, respectively. All three GnRH isoforms were sub-clustered with two other croaker species, large yellow croaker and Atlantic croaker, in each corresponding clade. However, three GnRH groups were supported by high bootstrap values: 100% for GnRH3, 99% for GnRH2, and 80% for GnRH1.

### 3.5. Synteny Analysis

After analyzing possible evolutionary connections of small yellow croaker GnRH genes using phylogenetic analysis, a synteny analysis was further performed to determine the origin and orthology relationship between each GnRH isoforms of small yellow croaker with four other teleost species and humans. The conserved synteny of syc-GnRH1, syc-GnRH2, and syc-GnRG3 locus was found on scaffold 717s1, 38, and 60 in the genome sequences of small yellow croaker, respectively (Figure 5), indicating that these three GnRH isoforms had their own syntenies. Small yellow croaker GnRH1 was flanked by KCTD9 and NPY8br on the left side and the right side, respectively, which were also present in all teleost species included in the present analysis. After KCTD9, GnRH1 neighboring with ANKRD, FGFR1A, and LGI3 on the left side. After NPY8br, it was neighboring with TBX16, NEFL, and DOCKS on the right side (Figure 5A). Thus, all teleost GnRH1 genes examined in the present analysis were orthologous to one another. However, in humans, KCTD9 was present on the left side of GnRH1 but DOCK5 present on the right side instead of NPY8br.

GnRH2 were also positioned in genomic regions containing common neighboring genes in most examined teleost species. Neighboring genes included ZMAT1, CD8B, CAST, and PCSK1 on the right side of GnRH2 and PTPRA, IDH3B, WDR1, and SLC2A9 on the left side of GnRH2 (Figure 5B). In humans, only PTPRA genes were neighbored on the left side of GnRH2; no other neighboring genes matched with any teleost species on either the right or the left side. In case of GnRH3, it was also positioned within a common gene cluster on both sides in all examined teleost genomes. Common neighboring genes of GnRH3 included MGMT, EBF3A, PPP2R, BNIP3 on the right side of GnRH3 and PTPRE, INSYN2A, ADAM12, and DHX32B on the left side of GnRH3 (Figure 5C). In humans, GnRH3 was absent although an almost similar neighboring gene arrangement was observed in the human genome.

### 3.6. Changes in mRNA Expression Levels of Three GnRH Isoforms in Different Organs of Small Yellow Croaker in Both Sexes

Tissue distribution analysis revealed that the small yellow croaker GnRH1, GnRH2, and GnRH3 genes were mainly expressed in the brain (Appendix A). However, GnRH1 also expressed in pituitary, significantly lower than the brain and significantly higher than other tissues. All three GnRHs were also expressed in different organs in negligible levels, which is common in teleost species [14,15,26].

### 3.7. Changes in mRNA Expression Levels of Three GnRH Isoforms in Different Brain Parts of Small Yellow Croaker in Both Sexes

Results of relative mRNA expression levels of syc-GnRH1, syc-GnRH2, and syc-GnRH3 in different brain area of fully mature small yellow croaker are presented in Figure 6. Different brain parts are illustrated in Figure 6D.

A differential mRNA expression of three GnRH isoforms were observed in the brain. Although females showed lower expression than males, these changes were insignificant. Both syc-GnRH1 and syc-GnRH3 were expressed in three examined brain areas, whereas expression of syc-GnRH2 was only limited in the midbrain region (Figure 6B). syc-GnRH1 expression was significantly higher in the TEL+POA area of the brain than in the OB and MB (Figure 6A). However, syc-GnRH3 showed significantly higher expression in OB than in the other two brain areas where its expression levels were very low (Figure 6C).

### 3.8. Changes in mRNA Expression Levels of Three GnRH Isoforms in Brain of Small Yellow Croaker during Gonadal Developmental Stages

The mRNA expression levels of syc-GnRH1 (sbGnRH) in the brain were significantly different in different gonadal developmental stages of both sexes (Figure 7A). The expression was lower during immature and spent stage, and the differences between these two stages were insignificant. However, its mRNA expression was significantly higher during the ripen stage compared to other stages in both sexes. Interestingly, females showed significantly lower expression during the ripen stage than males. The changes of expression of GnRH1 in females during DS and RS were insignificant. In case of syc-GnRH2 (cGnRH-II), mRNA expression levels did not show any significant differences among gonadal developmental stages of either sex (Figure 7B). In contrast, the expression levels of syc-GnRH3 (sGnRH) were increased at the ripen stage but dropped at the spent stage in females, although the changes were not significant. On the other hand, it was gradually increased in males from the immature stage to the spent stage, and significantly higher expressions were observed at the spent stage compared to other developmental stages. However, differences in mRNA expression between the ripen and spent stages were not significant (Figure 7C).

### 3.9. Changes in mRNA Expression Levels of Three GtH Subunits in Pituitary of Small Yellow Croaker during Gonadal Developmental Stages

The mRNA expression levels of all three GtH subunits in pituitary of females showed significantly lower expression than males in the ripen stage (Figure 8). The mRNA expression levels of GPα (Figure 8A), FSHβ (Figure 8B), and LHβ (Figure 8C) in males were increased significantly in DS and RS stages than in the IM stage. However, changes in mRNA levels were significant between DS and RS stages. Interestingly, expression levels of GtH subunits showed insignificant differences between DS and RS stage in females (Figure 8A–C). Expressions of all three GtH subunits were significantly downregulated in SS compared to in RS, and expression levels reached a level that was insignificant to IM.

### 3.10. Changes in mRNA Expression Levels of Three GnRH Isoforms in Brain of Small Yellow Croaker during Induced Spawning Event

The mRNA expression levels of syc-GnRH1 and syc-GnRH3 in brain were significantly higher at DSW than those in BSW and PSW stage in both sexes (Figure 9A,C). However, sycGnRH2 did not show any significant changes during spawning events (Figure 9B). Interestingly, expression levels of syc-GnRH1 in females showed significantly lower expression than males at the DSW stage (Figure 9A).

### 3.11. Time- and Dose-Dependent Effect of GnRH1 on mRNA Expression of Three GtH Subunits in Pituitary after In Vivo Injection of GnRH1 Peptide

The effects of GnRH1 peptide on mRNA expression of three GtH subunits (FSHβ, LHβ, and GPα) in the pituitary are illustrated in Figure 10. All three GtH subunits showed significant changes in mRNA expression in the pituitary after in vivo administration of GnRH1 peptide in a time- and dose-dependent manner. The mRNA expression levels of GPα were gradually increased until 6 h of post-injection and decreased at 12 h of post-injection. However, these increases were significantly higher than those of the initial control. Although the expressions levels of GPα were significantly increased at all doses of GnRH1 peptide compared to the initial control, the maximum increase was found at higher doses (Figure 10A). FSHβ showed significantly higher mRNA expression at a higher dose of GnRH1. However, it was significantly increased at 3 h of post-injection and decreased gradually till 12 h (Figure 10B). Similarly, mRNA expression levels of LHβ showed a significant increase in a higher dose of GnRH1, whereas a lower dose of GnRH1 peptide did not show any significant effect on the expression of LHβ. Expression levels of LHβ were also increased significantly at 3 h of post-injection and decreased at 12 h (Figure 10B).

### 3.12. Time- and Dose-Dependent Effect of GnRH1 on mRNA Expression of Three GtH Subunits in Cultured Pituitary after In Vitro Incubation with GnRH1 Peptide

Results of mRNA expression levels of three GtH subunits in pituitary after static culture of pituitary cells in the presence of different doses of GnRH1 peptide for 3, 6, and 12 h are summarized in Figure 11. It was observed that GnRH1 peptides at a lower dose had little stimulatory effect on all three GtH subunits. However, at higher doses (10 μM), expression levels of all three GtH subunits (GPα, FSHβ, and LHβ,) were increased significantly compared to that at the initial control (Figure 11A–C). In the time course study, expression levels of all three GtH subunits were found significantly higher at 6 h of post-incubation with 10 μM of GnRH1 peptide, and expression levels were downregulated at 12 h of post-incubation.

### 3.13. Time- and Dose-Depended Effect of 17β-Estradiol (E2) on mRNA Expression of Three GnRH Isoforms in Hypothalamus after In Vitro Incubation with E2

Time- and dose-dependent effects of E2 on mRNA expression levels of the three GnRH isoforms in the hypothalamus are summarized in Figure 12. Results showed that E2 inhibited the mRNA expression of syc-GnRH1 (sbGnRH) in a dose-dependent manner. Expression levels were significantly downregulated by all doses (0.1. 1, 10 μM) of E2 at all time points (3, 6, and 12 h) (Figure 12A). However, no effect was observed on a time course study from 3 to 12 h. On the other hand, E2 did not show any time- or dose-dependent effects on the expression of syc-GnRH2 or syc-GnRH3 (Figure 12B,C).

### 3.14. Time- and Dose-Depended Effect of 17α-Methyltestosterone (MT) on mRNA Expression of Three GnRH Isoforms in Hypothalamus after In Vitro Incubation with MT

Effect of MT on mRNA expression of the three GnRH isoforms in the hypothalamus at different time points and different doses of MT are presented in Figure 13. Similar to E2, MT also showed an inhibitory effect on the mRNA expression of syc-GnRH1 (sbGnRH) in a dose-dependent manner. However, syc-GnRH1 showed significantly lower expression at a higher dose (10 μM) of MT at 3 h, 6 h, and 12 h (Figure 13A). In contrast, it did not show any time- or dose-dependent effect on the expression of syc-GnRH2 or syc-GnRH3 (Figure 13B,C).

## 4. Discussion

Vertebrate reproduction is entirely dependent on the activity of gonadotropin-releasing hormone (GnRH) secreted from the brain in the BPG axis [2]. It is evident that vertebrates possess at least two GnRH isoforms in the brain. However, few teleost species possess three GnRH isoforms including Perciform fish. In the present study, full-length cDNA sequences of three GnRH isoforms encoding sbGnRH (syc-GnRH1), cGnRH-II (syc-GnRH2), and sGnRH (syc-GnRH3) were sequenced and characterized from brain tissue of a perciform fish, small yellow croaker (*L. polyactis*). These cloned three GnRH isoforms of small yellow croaker showed similar sequence features as those observed in other known GnRH isoforms. The presence of a classical conserved GnRH decapeptide in deduced amino acid sequences of three GnRH isoforms confirmed the corresponding GnRH isoforms. GO analysis also confirmed these genes as GnRH activity protein.

Amino acid sequence alignment revealed that small yellow croaker sbGnRH, cGnRH-II, and sGnRH aligned with other teleost GnRH1, GnRH2, and GnRH3 isoforms, respectively (Figure 2). Phylogenetic analysis shown that small yellow croaker GnRH1 (sbGnRH) isoform clustered with other known teleost GnRH1 isoforms, suggesting that small yellow croaker sbGnRH might be a homolog of GnRH1 isoform. It might have a close relationship with its closely related perciform species such as large yellow croaker and Atlantic croaker. However, syc-GnRH2 (cGnRH-II) and syc-GnRH3 (sGnRH) isoforms clustered with other teleost GnRH2 and GnRH3, respectively. The organization of three clades of the three GnRH isoforms of small yellow croaker in the phylogenetic tree supports an expected relationship based on the evolution of GnRHs in teleost [3]. The synteny analysis also confirmed that small yellow croaker possessed three paralogous GnRH isoforms, which were likely products of 2R genome duplication. The three GnRH isoforms of small yellow croaker had a syntenic match with GnRH isoforms of other teleost species such as large yellow croaker, yellowtail amberjack, fugu, and Japanese medaka. syc-GnRH1, syc-GnRH2, and syc-GnRH3 were flanked by genes encoding KCTD9 and NPY8br, PTPRA and ZMAT1, PTPRE, and MGMT, respectively, which are common genomic features in teleost GnRH isoforms [10]. The present synteny map constructed for three GnRH isoforms matched with the syntenic map reported previously for fish [9,10,36].

The three GnRH isoforms showed different mRNA expressions in discrete brain areas. syc-GnRH2 was exclusively expressed in the mid brain, consistent with available reports of all other species. syc-GnRH3 expressed in olfactory bulbs and syc-GnRH1 showed overlapping expression in olfactory bulb and telencephalon. Similar expression patterns of these three GnRH isoforms have also been reported in Atlantic croaker [37], European seabass [38], and Japanese anchovy [39]. It was found that the three GnRH isoforms of Japanese anchovy were differentially expressed in different brain regions and that GnRH1 showed an overlapping expression in OB and telencephalon [39] as observed in the present study.

In seasonal gonadal developmental stages, mRNA expression levels of syc-GnRH1 were significantly increased at the ripen stage in the brains of both sexes. On the other hand, mRNA levels of all three GtH subunits were increased simultaneously in the pituitary at the ripen stage of both sexes. Significantly higher levels of GnRH1 at the ripen stage have also been reported in several fish species such as turbot [26], Japanese anchovy [39], chub mackerel [40], and grass puffer [41]. In accordance with reference studies, results of the present study of higher mRNA expression of GnRH1 in the brain and GtH subunits in the pituitary at the ripen stage suggest that syc-GnRH1 might be involved in the gonadal maturation process of small yellow croaker by stimulating the secretion of GtHs in the pituitary. However, the expression levels of syc-GnRH1 in the female did not show significant differences among the developing and ripen stages. The expression of syc-GnRH1 at the ripen stage was significantly lower in the female than the male. Significantly lower expression of GnRH1 in the female compared to the male has also been reported in captive reared fish including turbot [26], jack mackerel [27], and chub mackerel [40]. The mRNA expressions of syc-GnRH2 in the brain of small yellow croaker did not show any significant changes among seasonal gonadal developmental stages. Insignificant changes of GnRH2 in brain and pituitary have also been reported in several fish species including chub mackerel [40] and grass puffer [41], suggesting that GnRH2 may not be directly involved in the secretion of pituitary GtHs or gonadal development. Rather, it may have neuromodulatory function in fish, such as playing a role in the modulation of pineal function and melatonin release [42] and sensory processing of sexual or communicative stimuli [43]. On the other hand, mRNA expression levels of syc-GnRH3 were increased gradually in the brain at different gonadal developmental stages in both sexes. However, these changes were not significant in females, but significantly higher expression was observed at the spent stage in males. Although its expression continued to show higher levels at the spent stage in males, it downregulated in females. A similar pattern of sexual dimorphic expression of GnRH3 has also been reported in a perciform fish, chub mackerel [40]. Sexual dimorphic expression of GnRH3 has also been observed in a tetradontiform fish, grass puffer [41], in which females showed a higher expression of GnRH3 in post spawning fish. Low and insignificant changes of GnRH3 in pituitary have also been reported in three GnRH expressing fish [40]. Insignificant changes in of GnRH3 in gonadal developmental stages may suggest that it is not involved in gonadal development or reproduction; rather, it may act as a neuromodulator [44], involved in the regulation of different spawning behaviour in fish, such as nest-building and aggressive behavior in male tilapia [45] and male dwarf gourami [46].

The mRNA expression levels of syc-GnRH1 and syc-GnRH3 were significantly increased during the spawning stage than in a stage before induction of GnRHa. The expressions were significantly downregulated during the post-spawning period. Although the expression of syc-GnRH1 increased significantly at spawning time, the expression was significantly lower in the female compared to the male. A previous study on grass puffer has shown that spawning stage puffer fish expressed higher mRNA levels of GnRH1 and GnRH3, whereas these levels were downregulated at the post-spawning season in the wild [41]. Similar results have also been observed in culture turbot [26]. However, in three-spined stickleback, a fish expressing two GnRH isoforms, mRNA expression levels of GnRH3 were significantly elevated in spawning fish but downregulated in post-spawning fish [36], where GnRH3 acts as GnRH1 [47]. It is believed that reproductive development and gonadal recrudescence are controlled by top-down stimulation of the BPG axis via GnRHs and GtHs with hypothalamic sensitivity to sex steroids providing negative feedback, respectively [48]. It has also been demonstrated that the capacity of high-affinity GnRH binding sites (presumably representing receptors) might be higher in the breeding season fish than that in the post-breeding season fish [49]. This may enhance the higher expression of GnRHs during breeding time in small yellow croaker.

GtHs (GPα, FSHβ, and LHβ) are glycoprotein hormones that produced in the pituitary. These hormones play a critical role in the BPG axis by regulating gonadal maturation in vertebrate including fish [11]. In the present study, mRNA expression levels of three GtH subunits were significantly increased during ripening stage in both sexes. Interestingly, expression of GtHs subunits was significantly lower in the female compared to the male at the ripen stage. Levels of mRNA expression of GtH subunits did not show any significant differences among the developing stage and ripen stage in females. Seasonal changes in mRNA expression of GtH subunits in pituitary of fish show different expression patterns [11]. In captive reared female jack mackerel, expressions levels of three GtH subunits did not show significant differences during ovarian development, and it was significantly lower compared to wild jack mackerel [27]. In cultured female chub mackerel, expression levels of FSHβ and LHβ were increased during ovarian development and reached a peak during vitellogenesis at the ripen stage [50]. Expression levels of the three GtH subunits were significantly higher in fully mature rainbow trout of both sexes [51]. On the other hand, it has been shown that FSHβ may play an important role during gametogenesis in males, but not in females, whereas LHβ may be involved in the regulation of both early and late gametogenesis of red seabream in both sexes [52]. The present study showed a similar result to what was observed in jack mackerel, chub mackerel, and rainbow trout, suggesting that these three GtH subunits might be involved in the regulation of the synthesis of sex steroids and gonadal maturation of small yellow croaker. Furthermore, lower expression of GtH subunits, especially LHβ in females compared to males might be responsible for reproductive dysfunction in female small yellow croaker. To further confirm this finding, levels of circulating GtHs and steroid need to be measured in future studies.

GnRHs regulate the synthesis and secretion of GtH subunits in the pituitary of fish [2]. It has been reported that GnRHs can differentially stimulate the expression of FSHβ and LHβ across fish species [11]. In the present study, it was observed that in vivo administration of GnRH1 peptides significantly changed the expression levels of three GtH subunits in a time- and dose-dependent manner in the pituitary of small yellow croaker. Similar expression patterns of FSHβ and LHβ after in vivo administration of GnRH peptides have been reported in spotted scat [14], pompano [15], striped bass [12] and three GtH subunits in black porgy [13], and goldfish [53]. Furthermore, in vitro incubation of pituitary with GnRH1 peptide showed that GnRH1 could stimulate all three GtH subunits at a higher dose at 6 h of post-incubation. In vitro results of the present study were aligned with those found in spotted scat [14], pompano [15], Mediterranean seabass [54], and coho salmon [55]. These in vivo and in vitro experiments of GnRH1 on expression of GtH subunits suggest that GnRH1 peptide could regulate the release of GtH subunits in a time- and dose-dependent manner. Therefore, it is important to find out and use an optimum dose of GnRH1 peptide or GnRH analogues during induced breeding of small yellow croaker in hatcheries.

GtH subunits can exert their action to produce steroid hormones (estradiol, E2 and testosterone, T) that in turn facilitate gametogenesis [50]. Once secreted, E2 and T levels can in turn downregulate BPG axis activity by binding with sex steroid hormone receptors to induce negative feedback, thus suppressing further hypothalamic and pituitary secretion [48,56]. The feedback regulation of sex steroids on hypothalamic GnRH has been well-documented in mammals and fish. In the present study, both E2 and MT significantly downregulated the expression of syc-GnRH1 in in vitro cultured brain in a time- and dose-dependent manner. However, these steroids did not show any effect on syc-GnRH2 or syc-GnRH3. The present results suggest that sex steroids E2 and MT employ negative feedback on syc-GnRH1 which is the hypothalamic form of GnRH and responsible for the release of GtHs. These results also imply that E2 and MT may directly regulate the synthesis of GnRH1. The negative feedback regulation of sex steroid as observed in the present study has also been reported in few teleost species including spotted scat [14] and pompano [15]. However, the negative effect of E2 on GnRH1 was eliminated in spotted scat by a broad-spectrum ERα-specific antagonist, suggesting that ERα is one of the gene responsible for the negative feedback of E2 [14]. Besides the negative feedback of steroids on GnRHs, positive feedback was also demonstrated in some fish species including tropical damselfish [57] and black porgy [58]. Therefore, more intense study is necessary on feedback regulation of sex steroids on GnRHs and GtHs as well.

In the present study, it was observed that the expressions of syc-GnRH1 in the brain and GtH subunits in the pituitary showed significantly lower levels at the ripen stage during gonadal development in the female small yellow croaker compared to the male. Furthermore, the changes of mRNA expression of syc-GnRH1 in the brain and expression of GtH subunits in the pituitary among DS and RS did not show significant differences in females. These results might mediate the reproductive dysfunction in female during final maturation. It has been reported that low levels of GnRH1 and GtH subunits, especially LHβ, causes reproductive dysfunction i.e., failed to undergo complete final maturation in female chub mackerel maintained in captivity [40,50]. Lower levels of GnRH1 expression also reported in females compared to males in several fish including culture turbot [26], captive-reared jack mackerel [27], and long-whiskered catfish [28] that lead to reproductive dysfunction in captive reared females of respective fish.

## 5. Conclusions

In conclusion, three GnRH isoforms were cloned from small yellow croaker in this study to elucidate key information about the potential role of GnRHs on the BPG axis of captive-reared small yellow croaker. Three GnRH isoforms were distinctly expressed in three discrete brain areas. The mRNA expression of syc-GnRH1 was significantly upregulated in ripened fish and spawning time as well. Expression levels of the three GtH subunits were simultaneously upregulated in ripened fish. However, expression of GnRH1 and three GtH subunits in females showed significantly lower expression during the ripen stage in the female compared to the male. Furthermore, in vivo and in vitro administration of GnRH1 significantly elevated the expression of the three GtH subunits in a time- and dose-dependent manner. The feedback regulation of sex-steroids in vitro on GnRH expression revealed that E2 and MT could mediate negative feedback regulation on syc-GnRH1 expression. To further confirm this finding, future research is needed on circulating levels of GtHs and sex steroids in developmental stages. Taken together, present results suggest that syc-GnRH1 plays a central role in the regulation of BPG axis, which might regulate gonadal maturation through a stimulatory effect on the secretion of GtHs, and sex steroids might play a negative feedback role on syc-GnRH1. Finally, lower expression levels of syc-GnRH1 and GtH subunits, particularly LHβ in the brain and pituitary, respectively, in females compared to males, might be responsible for the reproductive dysfunction in the female small yellow croaker reared in captivity.

## Figures and Tables

**Figure 1 biology-11-01200-f001:**
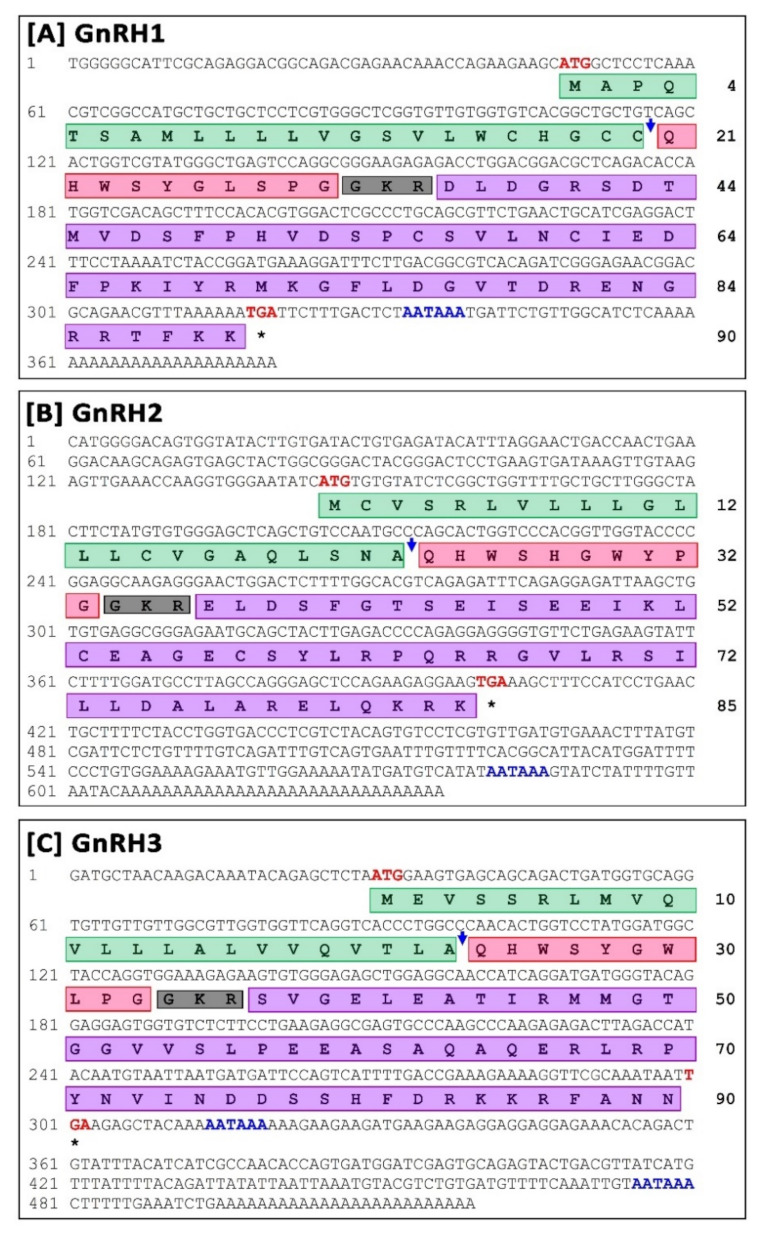
Full-length nucleotide and deduced amino acid sequences of small yellow croaker GnRH1 (**A**), GnRH2 (**B**), and GnRH3 (**C**). Numbers on the left and right side indicate nucleotide and amino acid positions in the sequences, respectively. Start and stop codons are marked in bold red. The stop codon is also indicated by an asterisk (*). Putative polyadenylations signals (AATAAA) are shown in bold blue. Putative signal peptides are boxed in green, GnRH decapeptide regions are highlighted in red box, cleavage sites are boxed in black, and GnRH-associated peptides are boxed in purple. GenBank nucleotide accession numbers and protein IDs are OK042349 and UFT26658.1 (syc-GnRH1), OK042350 and UFT26659.1 (syc-GnRH2), OK042351, and UFT26660.1 (syc-GnRH3).

**Figure 2 biology-11-01200-f002:**
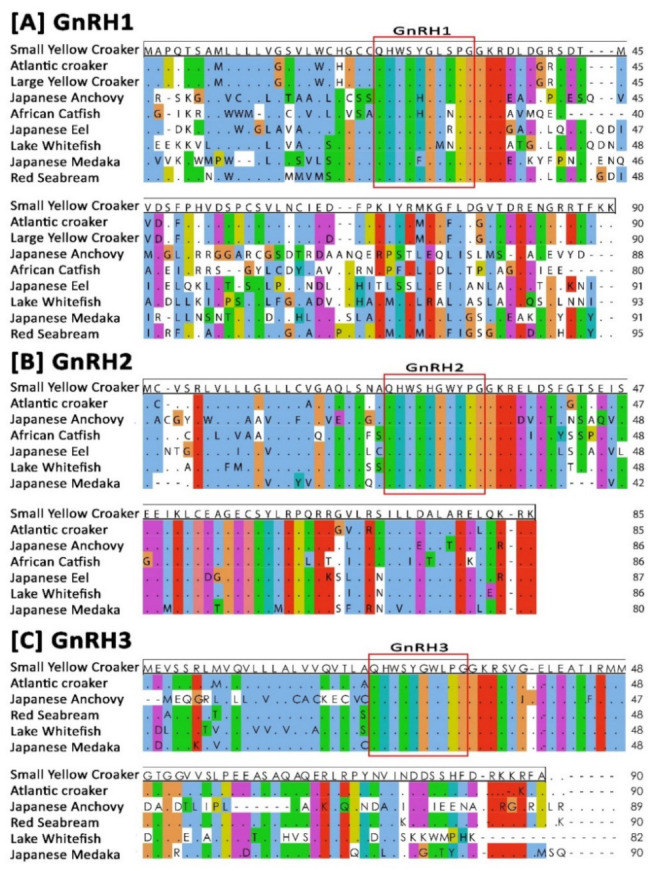
Multiple sequence alignment of deduced amino acid sequences of GnRH1 (**A**), GnRH2 (**B**), and GnRH3 (**C**). Identical amino acids are indicated by dots. Dashes indicate gaps. Corresponding amino acid sequences of GnRH decapeptides are marked in a red box. GenBank accession numbers of each GnRH isoforms are presented in Appendix A.

**Figure 3 biology-11-01200-f003:**
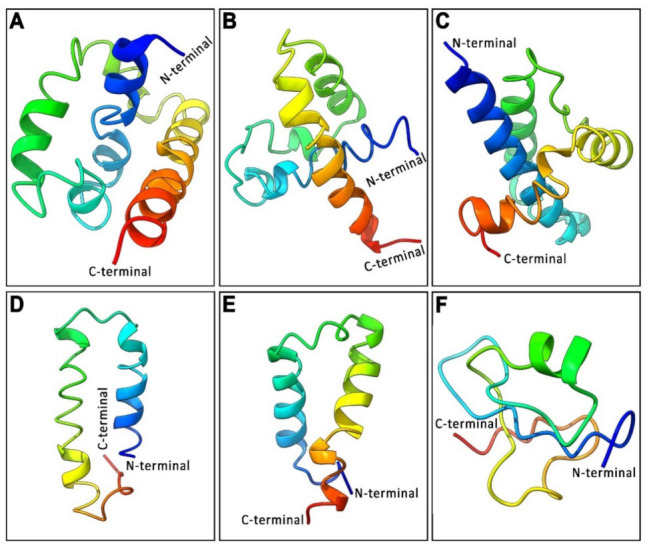
Predicted 3D structures of the three GnRH isoforms and corresponding three GnRH associated peptide (GAP) of small yellow croaker: (**A**) syc-GnRH1, (**B**) syc-GnRH2, (**C**) syc-GnRH3, (**D**) syc-GAP1, (**E**) syc-GAP2, and (**F**) syc-GAP3. The model was constructed using an I-TASSER online tool. Domains between the N-terminal and C-terminal were predicted from the secondary structure.

**Figure 4 biology-11-01200-f004:**
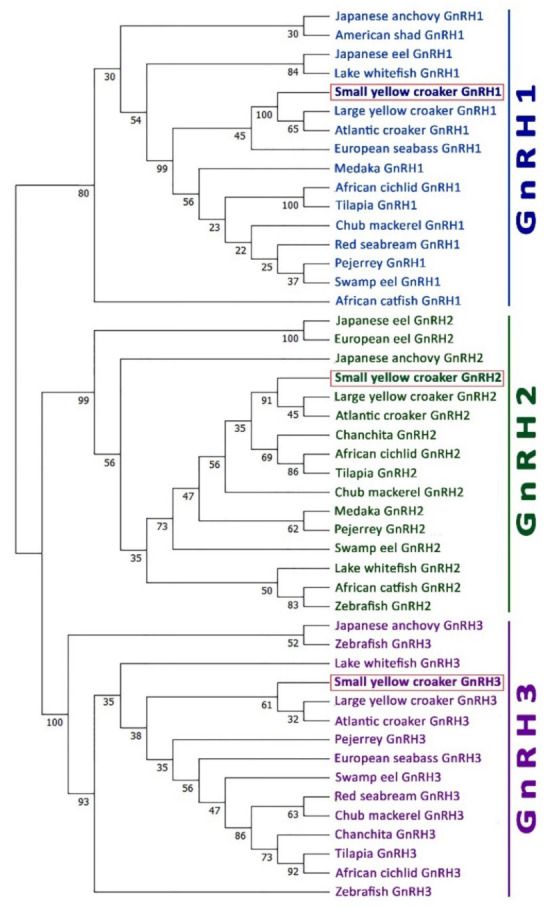
Phylogenetic tree of teleost GnRH isoform sequences. Phylogenetic analysis of 47 teleost GnRH amino acid sequences were performed by the bootstrap neighbor-joining method with 10,000 bootstrap replicates after aligning amino acid sequences of different teleost GnRH isoforms using the MUSCLE alignment option. Numbers at the nodes indicate bootstrap probability. GenBank accession numbers of GnRH sequences used to construct the phylogenetic tree are presented in Appendix A.

**Figure 5 biology-11-01200-f005:**
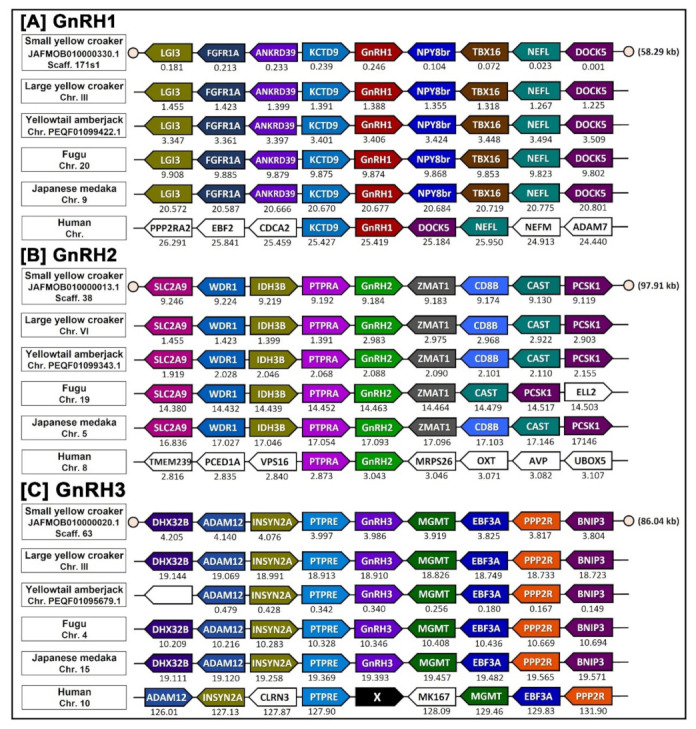
Synteny map comparing orthologs of GnRH1 (**A**), GnRH2 (**B**), and GnRH3 (**C**) locus and genes flanking GnRH1, GnRH2, and GnRH3, respectively, in the small yellow croaker and four selected teleost species (large yellow croaker, yellowtail amberjack, fugu, and Japanese medaka) and humans. The map was constructed based on results obtained from the genome browser Gnenomicus v. 106. Genes are represented by block arrows. Orthologs of each gene in different species are shown in the same column and colored. Position of the gene (megabases, Mb) is displayed below each block arrow according to the Ensembl database. Empty circles indicate the end of scaffolds. Detailed chromosomal locations of genes displayed in this map are presented in Appendix A.

**Figure 6 biology-11-01200-f006:**
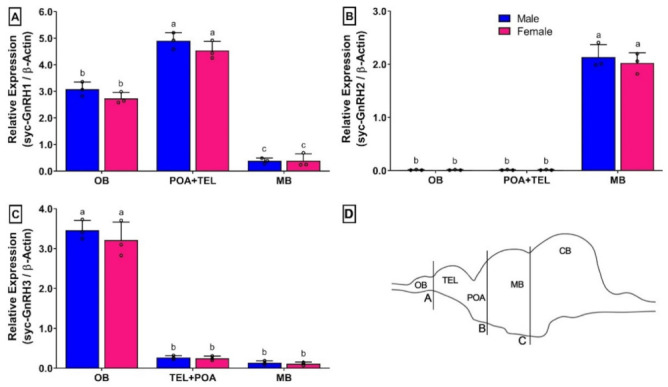
Relative mRNA expression levels (2^−ΔΔCT^) of three GnRH isoforms in different brain areas of small yellow croaker detected by qRT-PCR: (**A**) syc-GnRH1, (**B**) syc-GnRH2, and (**C**) syc-GnRH3, and (**D**) schematic diagram of brain part. For all bar graphs, raw data points (black circle) represent biological replicates, error bars represent standard deviation, and different letters above the bars indicate significant differences (*p* < 0.05) among organs. OB, olfactory bulb; TEL, telencephalon; POA, preoptic area; MB, midbrain.

**Figure 7 biology-11-01200-f007:**
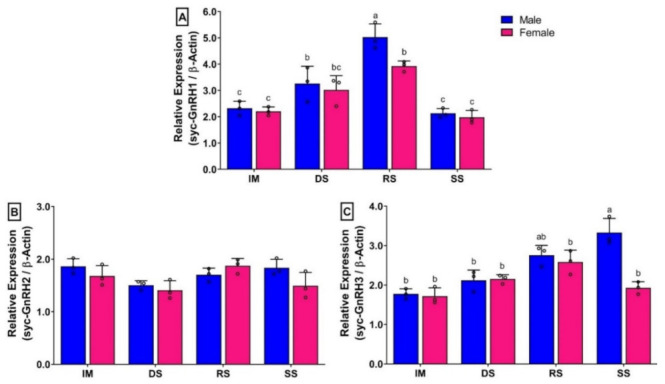
Relative mRNA expression levels (2^−ΔΔCT^) of three GnRH isoforms in the brain of small yellow croaker during gonadal developmental stages in males and females reared in captivity: (**A**) syc-GnRH1, (**B**) syc-GnRH2, and (**C**) syc-GnRH3. For all bar graphs, raw data points (black circle) represent biological replicates, error bars represent standard deviation, and different letters above the bars indicate significant differences (*p* < 0.05) among developmental stages. IM, immature; DS, developing stage; RS, ripen stage; SS, spent stage.

**Figure 8 biology-11-01200-f008:**
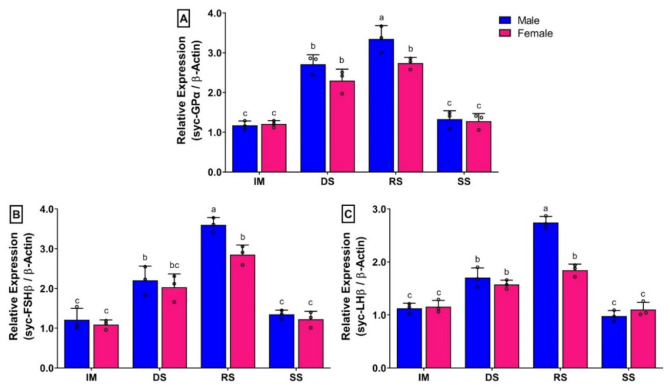
Relative mRNA expression levels (2^−ΔΔCT^) of three GtH subunits in the pituitary during gonadal development stages in captive reared male and female small yellow croaker: (**A**) GPα, (**B**) FSHβ, and (**C**) LHβ. For all bar graphs, raw data points (black circle) represent biological replicates, error bars represent standard deviation, and different letters above the bars indicate significant differences (*p* < 0.05) among developmental stages. IM, immature; DS, developing stage; RS, ripen stage; SS, spent stage.

**Figure 9 biology-11-01200-f009:**
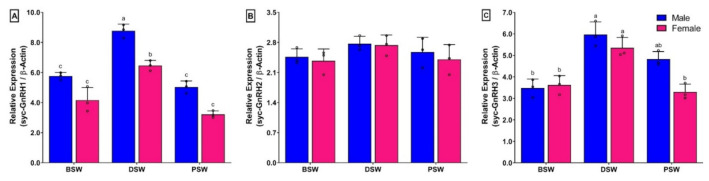
Relative mRNA expression levels (2^−ΔΔCT^) of three GnRH isoforms in the brain during induced spawning events in captive reared male and female small yellow croaker: (**A**) syc-GnRH1, (**B**) syc-GnRH2, and (**C**) syc-GnRH3. For all bar graphs, raw data points (black circle) represent biological replicates, error bars represent standard deviation, and different letters above the bars indicate significant differences (*p* < 0.05) among spawning events. BSW, before spawning; DSW, during spawning; PSW, post spawning.

**Figure 10 biology-11-01200-f010:**
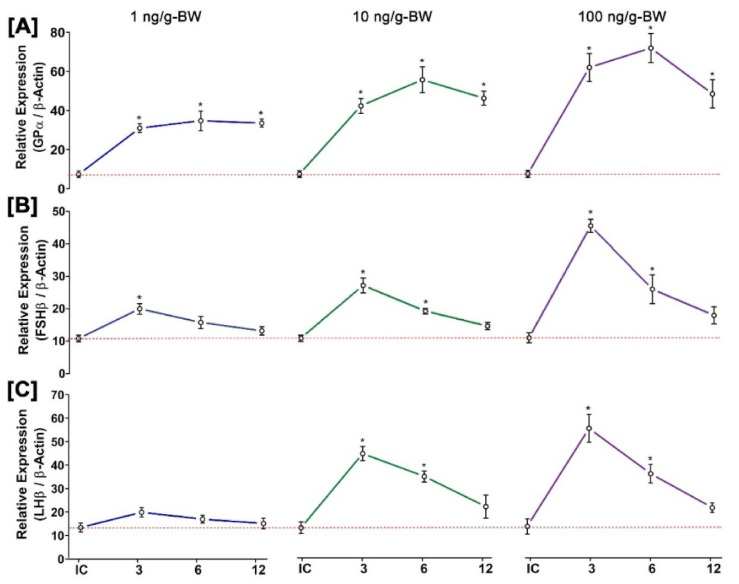
Time- and dose-dependent in vivo effects of GnRH1 peptide on mRNA expression levels of three GtH subunits in the pituitary of small yellow croaker: (**A**) GPα, (**B**) FSHβ, and (**C**) LHβ. GnRH1 peptide was injected at concentrations of 1, 10, and 100 ng/g-BW. Asterisks indicate significant differences (*p* < 0.05) compared to zero time point initial control (dotted lines).

**Figure 11 biology-11-01200-f011:**
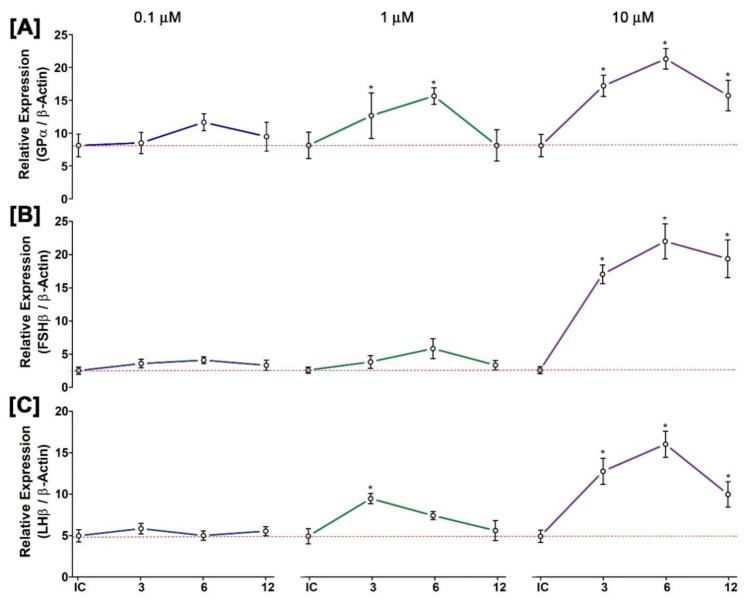
Time- and dose-dependent in vitro effects of GnRH1 peptide on mRNA expression levels of three GtH subunits in cultured pituitary of small yellow croaker: (**A**) GPα, (**B**) FSHβ, and (**C**) LHβ. GnRH1 peptide was applied in the culture media at concentrations of 0.1, 1, and 10 μM. Asterisks indicate significant differences (*p* < 0.05) compared zero time point initial control (dotted lines).

**Figure 12 biology-11-01200-f012:**
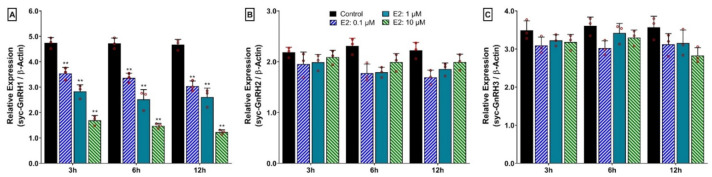
Time- and dose-dependent in vitro effects of 17-β estradiol (E2) on mRNA expression levels of three GnRH isoforms in cultured hypothalamus of small yellow croaker: (**A**) syc-GnRH1, (**B**) syc-GnRH2, and (**C**) syc-GnRH3. E2 was applied in the culture media at concentrations of 0.1, 1, and 10 μM. For all bar graphs, raw data points (red circle) represent biological replicates, error bars represent standard deviation, and asterisks indicate significant differences (*p* < 0.05) compared to the zero time point initial control.

**Figure 13 biology-11-01200-f013:**
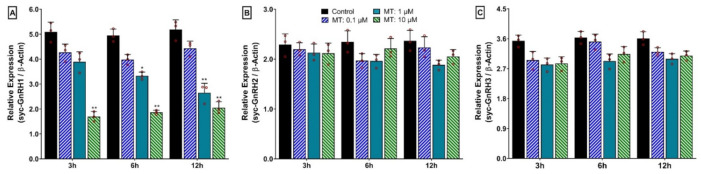
Time- and dose-dependent in vitro effects of 17-α methyltestosterone (MT) on mRNA expression levels of three GnRH isoforms in cultured hypothalamus of small yellow croaker: (**A**) syc-GnRH1, (**B**) syc-GnRH2, and (**C**) syc-GnRH3. MT was applied in the culture media at concentrations of 0.1, 1, and 10 μM. For all bar graphs, raw data points (red circle) represent biological replicates, error bars represent standard deviation, and asterisks indicate significant differences (*p* < 0.05) compared to the zero time point initial control.

## Data Availability

Not applicable.

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
