# Peer review of "Functional Characterization of Three GnRH Isoforms in Small Yellow Croaker Larimichthys polyactis Maintained in Captivity: Special Emphasis on Reproductive Dysfunction"

_biology, 2022, doi:10.3390/biology11081200_

Round 1

Reviewer 1 Report

The authors did a very extensive and thorough study. I don't have any great suggestions to make. However, in order to make the reading less tiring, I suggest that the Introduction be shortened.

Perhaps, the discussion could be more concise too; or the authors can add subheadings to the discussion to lighten the reading.

Small points:

1) What phase is “gonadal recovery phase”? Is it regressed, regeneration? How do the authors know that the specimens were at this stage?

2) How many animals were used in total? 10?

3) How do the authors know the gonadal stages (phases) of fish? If the classification was made only by macroscopic analysis, they have to be careful. Often, the microscopic analysis indicates errors in the classification by macroscopic observation.

4) Please, change “were sacrificed” to “were euthanized" (line 164).

5) What characters were used in phylogenetic analysis?

After the authors make these minor changes, I suggest that the manuscript be accepted for publication.

Author Response

Response to Reviewer 1 Comments

Point 1: The authors did a very extensive and thorough study. I don't have any great suggestions to make. However, in order to make the reading less tiring, I suggest that the Introduction be shortened.

Response 1: Thank you for your valuable suggestion. We have shortened the introduction section according to your suggestion. We have removed almost 10 lines from the introduction and rewritten several sentences where necessary.

Point 2: Perhaps, the discussion could be more concise too; or the authors can add subheadings to the discussion to lighten the reading.

Response 2: According to your suggestion, we have concise the discussion section.

Small points:

Point 3:  What phase is “gonadal recovery phase”? Is it regressed, regeneration? How do the authors know that the specimens were at this stage?

Response 3: It was in regeneration phase. With the annual gonadal histological study, we pointed out this phase and other gonadal developmental stages as well. In hatchery, they determine it according to monthly histological data and specific time period.

Point 4:  How many animals were used in total? 10?

Response 4: We used 10 fish in cDNA cloning and tissue distribution experiment, sampled 10 fish in each developmental stage (total 40 fish); 18 fish in spawning events. For in vivo and in vitro experiment 30 fish were used for each experiment (total 90 fish).

Point 5:  How do the authors know the gonadal stages (phases) of fish? If the classification was made only by macroscopic analysis, they have to be careful. Often, the microscopic analysis indicates errors in the classification by macroscopic observation.

Response 5: Gonadal developmental stages was determined using gonadal histology during maturation process. The stages were also confirmed following previously published paper.

Point 6:  Please, change “were sacrificed” to “were euthanized" (line 164).

Response 6: As suggested, “were sacrificed” was replaced with “were euthanized". [Line 155]

Point 7:  What characters were used in phylogenetic analysis?

Response 7: Phylogenetic tree was constructed using molecular data i..e, related protein sequences of GnRH isoforms form teleost species. The evolutionary history was inferred using the Neighbor-Joining method.  The evolutionary distances were computed using the Poisson correction method. 

Reviewer 2 Report

 Sukhan et al. cloned 3 forms of GnRHs in the small yellow croaker and characterised their expression in the different brain regions throughout gonadal development in both sexes. Pituitary GTH subunit expression was also determined at different stages. In vivo and in vitro experiments were performed to determine the effects of GnRH1 treatment and feedback of MT and E2 on GnRH1 expression.

 Overall, the study provides a good understanding of the GnRH system in the small yellow croaker. The authors are requested to modify their conclusions regarding the negative feedback effect of MT and E2 on GnRH1 because this was based on an in vitro experiment on hypothalamus from one stage only and the circulating levels of sex steroids were not determined. It is also suggested that authors modify their conclusion regarding the reproductive dysfunction in the small yellow croaker being attributed to low GnRH1 and GTH subunit levels as their data are contrary to this claim (see Figure 7a and Figure 8). The Discussion can be modified by saying that further studies measuring the circulating levels at least of GTHs would clarify this issue.

 There are minor language corrections, either grammatically or way of expression, that should be made throughout the manuscript. There are details in some sections of the methods that need to be added. The Discussion can be shortened. There is no need to repeat the results in the text.

 Below are further comments or guide for the authors:

Line 18 and 31 – Change “To know” to “determine” or “to confirm”.

 Line 39 – State that this was in vitro.

 Inappropriate choice of words in some instances; word cut incorrectly (lines 229-230; lines 269-270); line 506 ‘flanked’ would be more appropriate than ‘surrounded’.

 Section 2.4.6. Provide more detail – One pituitary per well? Separated pituitaries according to whether male or female?

 Section 2.4.7. State briefly the aim of this in vitro study. Were the hypothalamus incubated separately? Were the hypothalamus from male and female fish identified and incubated separately?

 Line 316 – What does AS represent?

 Section 2.12. Was there previous study proving that β-actin was a reliable internal reference for qRT-PCR in small yellow croaker?

 Section 3.2. Table 1 is just a repeat of Figure 2. Move Table 1 as Supplementary material.

 Line 608, 629 – dose-dependent not dose-depended.

 Line 827 – “GTH subunits can exert their action…” – This is not accurately stated.

Author Response

Response to Reviewer 2 Comments

Sukhan et al. cloned 3 forms of GnRHs in the small yellow croaker and characterized their expression in the different brain regions throughout gonadal development in both sexes. Pituitary GTH subunit expression was also determined at different stages. In vivo and in vitro experiments were performed to determine the effects of GnRH1 treatment and feedback of MT and E2 on GnRH1 expression.

Point 1: Overall, the study provides a good understanding of the GnRH system in the small yellow croaker. The authors are requested to modify their conclusions regarding the negative feedback effect of MT and E2 on GnRH1 because this was based on an in vitro experiment on hypothalamus from one stage only and the circulating levels of sex steroids were not determined.

Response 1: Thank you for your suggestion. We have rewritten the conclusion section and added following sentence “To further confirm this findings, future research is needed on circulating levels of GtHs and sex steroid in developmental stages.”

In discussion section, we have already mention in 1st draft of manuscript that “Therefore, more intense study is necessary on feedback regulation of sex steroids on GnRHs and GtHs as well.”

Point 2: It is also suggested that authors modify their conclusion regarding the reproductive dysfunction in the small yellow croaker being attributed to low GnRH1 and GTH subunit levels as their data are contrary to this claim (see Figure 7a and Figure 8).

Response 2: Thank you for your suggestion. Although the levels of GnRH1 and GtHs in female seems to be higher, the changes of expression between developing stage to ripen stage was insignificant. So that, female may not mature fully during spawning season. Further, the expression levels were much lower in female compared to male at ripen stage and it was significantly different. At this stage, male become fully mature but female not, and cannot undergo final maturation and vitellogenesis, and did not take part in spawning. Based on the significantly lower expression of GnRH1 and GtHs in female compared to male during ripen stage or spawning period, we concluded that lower expression of GnRH1 and GtHs in female compared to male may be responsible for reproductive dysfunction in female small yellow croaker. We made few changes in the conclusion section and discussion as well mentioning that lower expression levels in female compared to male.

Point 3: The Discussion can be modified by saying that further studies measuring the circulating levels at least of GTHs would clarify this issue.

Response 3: Thank you for your suggestion. We have added this statement in the revised manuscript. The sentence is “To further confirm this finding, levels of circulating GtHs and steroid need to measure in future studies.”

 Point 4: There are minor language corrections, either grammatically or way of expression, that should be made throughout the manuscript. There are details in some sections of the methods that need to be added.

Response 4: We have corrected some minor language correction throughout the manuscript as suggested.

Point 5: The Discussion can be shortened. There is no need to repeat the results in the text.

Response 5: According to your suggestion, discussion section have been shortened and some repeat results were deleted where applicable.

Below are further comments or guide for the authors:

Point 6: Line 18 and 31 – Change “To know” to “determine” or “to confirm”.

Response 6:  As suggested, “to know” was replaced with “to determine” in line 18 and 31.

Point 7: Line 39 – State that this was in vitro.

Response 7:   Phrase “in vitro” has been added in the sentence.

Point 8: Inappropriate choice of words in some instances.

Point 8-1: word cut incorrectly (lines 229-230; lines 269-270);

Response 8-1:  The word cut at lines 229-230 and lines 269-270 and other several cases were due to the format of journal template. In the title also a word cut at lines 3-4, this has been fixed in the revised manuscript.

Point 8-2: line 506 ‘flanked’ would be more appropriate than ‘surrounded’.

Response 8-2:  As suggested, “surrounded” was replaced with “flanked”.

Point 9: Section 2.4.6. Provide more detail – One pituitary per well? Separated pituitaries according to whether male or female?

Response 9:  No, three pituitaries were placed per well. As female small yellow croaker showed reproductive dysfunction, only female pituitaries were used for this in vitro study. We have included these issues in the revised manuscript. [Line 187, Line 190]

Point 10: Section 2.4.7. State briefly the aim of this in vitro study. Were the hypothalamus incubated separately? Were the hypothalamus from male and female fish identified and incubated separately?

Response 10:  As suggested, aim the in vitro has been included in the revised manuscript. Three hypothalamuses were incubated in a well. As female small yellow croaker showed reproductive dysfunction, only female hypothalamus was used for this in vitro study. We have included these issues in the revised manuscript.

Point 11: Line 316 – What does AS represent?

Response 11:  It was a typo mistake. It has been corrected in the revised manuscript as IM, DS, RS, SS.

Point 12: Section 2.12. Was there previous study proving that β-actin was a reliable internal reference for qRT-PCR in small yellow croaker?

Response 12:  Although limited references have been found on the molecular qRT-PCR analysis in small yellow croaker, there have few recent publications where β-actin was used as internal reference for qRT-PCR. Followings are the references

  1. Ma, B., Wang, L., Lou, B., Tan, P., Xu, D. and Chen, R., 2020. Dietary protein and lipid levels affect the growth performance, intestinal digestive enzyme activities and related genes expression of juvenile small yellow croaker (Larimichthys polyactis). Aquaculture Reports17, p.100403.
  2. Chu, T., Liu, F., Qin, G., Zhan, W., Wang, M. and Lou, B., 2020. Transcriptome analysis of the Larimichthys polyactis under heat and cold stress. Cryobiology96, pp.175-183..

Point 13: Section 3.2. Table 1 is just a repeat of Figure 2. Move Table 1 as Supplementary material.

Response 13:  As suggested, Table 1 has been moved to supplementary materials.

Point 14: Line 608, 629 – dose-dependent not dose-depended.

Response 14: “dose-depended” has been replaced with “dose dependent”

Point 15: Line 827 – “GTH subunits can exert their action…” – This is not accurately stated.

Response 15:  We have rewritten the sentence to make it more understandable and accurate.
